


# Long-term profiling of aerosol light-extinction, particle mass, cloud condensation nuclei, and ice-nucleating particle concentration over Dushanbe, Tajikistan, in Central Asia

Julian Hofer[1], Albert Ansmann[1], Dietrich Althausen[1], Ronny Engelmann[1], Holger Baars[1], Sabur F. Abdullaev[2], and Abduvosit N. Makhmudov[2]

[1]Leibniz Institute for Tropospheric Research, Leipzig, Germany
[2]Physical Technical Institute of the Academy of Sciences of Tajikistan, Dushanbe, Tajikistan

**Correspondence:** hofer@tropos.de

**Abstract.**

For the first time, continuous vertically resolved long-term aerosol measurements were conducted with a state-of-the-art multiwavelength lidar over a Central Asian site. Such observations are urgently required in efforts to predict future climate and environmental conditions and to support spaceborne remote sensing (ground truth activities). The lidar observations were per-

formed in the framework of the Central Asian Dust Experiment (CADEX) at Dushanbe, Tajikistan, from March 2015 to August 2016. An AERONET sun photometer was operated at the lidar field site. During the 18-month campaign, mixtures of continental aerosol pollution and mineral dust were frequently detected from ground to cirrus height level. Regional sources of dust and pollution as well as long-range transport of mineral dust mainly from Middle East and the Saharan deserts determine the aerosol conditions over Tajikistan. In this study, we summarize our findings and present seasonally resolved statistics regarding

aerosol layering (main aerosol layer depth, lofted layer occurrence), optical properties (aerosol and dust optical thicknesses at 500-532 nm, vertically resolved light-extinction coefficient at 532 nm), profiles of dust and non-dust mass concentration and dust fraction, and profiles of particle parameters relevant for liquid-water, mixed-phase cloud and cirrus formation such as cloud condensation nuclei (CCN) and ice-nucleating particle (INP) concentration. The main aerosol layer over Dushanbe reaches typically 4-5 km height in spring to autumn. Frequently lofted dust-containing aerosol layers were observed at heights

from 5-10 km, indicating a sensitive potential of dust to influence cloud ice formation. Typical dust mass fractions were of the order of 60–80%. A considerable fraction is thus anthropogenic pollution and biomass burning smoke. The highest aerosol pollution levels (in the relatively shallow winter boundary layer) occur during the winter months. The seasonal mean 500 nm AOT ranges from 0.15 in winter to 0.36 in summer during the CADEX period (March 2015 to August 2016), DOTs were usually below 0.2, seasonally mean particle extinction coefficients were of the order of 100–500 $\mathrm{Mm}^{-1}$ in the main aerosol layer

during the summer half year, and about 100-150 $\mathrm{Mm}^{-1}$ in winter, but mainly caused by anthropogenic haze. Accordingly, the highest dust mass concentrations occur in the summer season (200-600 $\mu\mathrm{g\,m}^{-3}$) and the lowest during the winter months (20-50 $\mu\mathrm{g\,m}^{-3}$) in the main aerosol layer. In winter, the aerosol pollution mass concentrations were 20-50 $\mu\mathrm{g\,m}^{-3}$, while during the summer half year (spring to autumn) the mass concentration caused by urban haze and biomass burning smoke decreases to 10-20 $\mu\mathrm{g\,m}^{-3}$ in the lower troposphere. The CCN concentration levels are always controlled by aerosol pollution. The INP



concentrations were found to be high enough in the middle and upper troposphere to significantly influence ice formation in mixed-phase and ice clouds during spring and summer seasons.

# 1 Introduction

Shrinking glaciers (Sorg et al., 2012, 2014; Ji et al., 2016; Farinotti et al., 2015; Kraaijenbrink et al., 2017; Schmale et al., 2017) and desiccating Aral Sea (Issanova et al., 2015; Li and Sokolik, 2017) are clear and unambiguous signs for major and threatening effects of human activities and climate change in Central Asia (Kazakhstan, Turkmenistan, Uzbekistan, Kyrgyzstan, Tajikistan, see Fig. 1). Aerosol pollution and mineral dust are important components in the environmental/atmospheric system in this region which belongs to the northern hemispheric dust belt extending from the Sahara in North Africa to the Taklamakan and Gobi deserts in China (Ginoux et al., 2012; Ridley et al., 2016; Hofer et al., 2017). Long-range transport of desert dust from the Sahara and the Middle East deserts and additional local and regional emissions of dust and aerosol pollution (anthropogenic haze, biomass burning smoke) lead to a complex aerosol mixture and complex vertical layering of aerosols in the planetary boundary layer and free troposphere over, e.g., Dushanbe in Tajikistan (Hofer et al., 2017).

Although Central Asia is a hot spot region of severe environmental problems and potentially dramatic climate-change effects, only a few observational studies on atmospheric aerosols were performed so far (e.g., Pachenko et al., 1993; Golitsyn and Gillette, 1993; Chen et al., 2013) and with focus on Tajikistan by Abdullaev and Sokolik (2019). First systematic characterization of atmospheric aerosol/pollution/dust conditions in terms of maps of aerosol optical thickness AOT, and Ångström exponent AE (describing the spectral dependence of AOT) for Central Asia were recently presented by Li and Sokolik (2018) and Rupakheti et al. (2019). AOT is a proxy for the tropospheric aerosol burden in the vertical column and AE can be used to identify and separate dust and non-dust fine-mode aerosol pollution fractions in the observed aerosol mixtures. These two studies are based on well-established methods of passive remote sensing from space. However, passive remote sensing does not allow us to adequately resolve the vertical aerosol structures as needed in state-of-the-art environmental and climate research and modeling efforts (Wiggs et al., 2003; Yorks et al., 2009; Lioubimtseva and Henebry, 2009; Huang et al., 2014; Xu et al., 2016; Bi et al., 2016; Kipling et al., 2016; Kok et al., 2018; Shi et al., 2019). Active remote sensing with ground-based and spaceborne lidars is required to provide the missing information on dust plume heights, long-range dust transport features, and to detect even thin dust layers in the upper troposphere which influence cloud and precipitation formation (Creamean et al., 2013; Ansmann et al., 2019a, b). Liu et al. (2008), Marinou et al. (2017), and Georgoulias et al. (2018) provide height-resolved dust climatologies based on active remote sensing from space with lidar aboard the CALIPSO (Cloud-Aerosol Lidar and Infrared Pathfinder Satellite Observations) satellite. These studies focused on main dust source regions such as North African, Middle East, and East Asian dust source regions. Such a dust profile climatology is however still missing for Central Asia.

Motivated by these observational gaps, we deployed a multiwavelength polarization/Raman aerosol lidar at Dushanbe (38.6°N, 68.9°E, 864 m height above sea level, a.s.l.), Tajikistan, in the framework of the CADEX (Central Asian Dust Experiment) project. The lidar (Polly: POrtabLe Lidar sYstem) (Engelmann et al., 2016; Baars et al., 2016) was continuously operated over a 18-month period from March 2015 to August 2016. First results were presented by Hofer et al. (2017). Profiles of basic



aerosol optical properties and dust mass concentration in combination with vertically resolved dust source identification for representative aerosol scenarios were discussed based on case studies. The final results of this campaign are presented in this article (in Sect. 3). More than 300 individual (day by day) nighttime observations are analyzed and cover well the annual cycle of dust and aerosol pollution layering. As a follow-up project, we recently build a containerized Polly instrument and deployed this new lidar at Dushanbe (June 2019) for long term observations over the next 5-10 years. In addition, we organized the first Central Asian Dust Conference (CADUC 2019, 8-12 April 2019) to emphasize the importance of Central Asian pollution and dust in the global climate system and need for more research in this region (Althausen et al., 2019).

The article is structured as follows: In Sect. 2, we briefly provide technical details to the Polly lidar and the data analysis. The POLIPHON (Polarization Lidar Photometer Networking) data analysis scheme (Mamouri and Ansmann, 2016, 2017; Ansmann et al., 2019b) was applied to derive aerosol-type-dependent particle optical properties, dust and non-dust (haze, smoke) mass concentrations profiles, and cloud-process-relevant aerosol parameters such as cloud condensation nucleus (CCN) and ice-nucleating particle (INP) concentrations. In Sect. 4, the main findings are discussed and summarized. Concluding remarks are given in Sect. 5.

## 2 CADEX lidar data analysis

During the 18-month CADEX campaign a Polly-type multiwavelength polarization/Raman lidar (Althausen et al., 2009; Engelmann et al., 2016; Hofer et al., 2017) was operated in Dushanbe, Tajikistan. The Dushanbe lidar station is part of PollyNET, a network of permanent or campaign-based Polly lidar stations (Baars et al., 2016) and is the first outpost of the European Aerosol Research Lidar Network (EARLINET) (Pappalardo et al., 2014). The polarization Raman lidar permits us to measure height profiles of the particle backscatter coefficient at the laser wavelengths of 355, 532 and 1064 nm wavelength, particle extinction coefficients at 355 and 532 nm by means of 387 and 607 nm nitrogen Raman signal profiling, the particle linear depolarization ratio at 355 and 532 nm by means of additional cross-polarized lidar return detection at 355 and 532 nm, and of the water-vapor-to-dry-air mixing ratio by using the Raman lidar return signals at 407 nm (water vapor channel) and 387 nm nitrogen Raman channel (e.g., Mattis et al., 2004; Baars et al., 2012; Engelmann et al., 2016; Hofer et al., 2017; Dai et al., 2018). Technical details of the lidar system are described in Engelmann et al. (2016). The specifically used Polly, the field site and the CADEX measurement campaign including ancillary instrumentation are described in Hofer et al. (2017).

The lidar observations were manually analyzed. During the 18-month CADEX campaign (535 days), the Polly lidar acquired data at 487 days for at least a 3 h time period. To achieve a representative coverage of aerosol conditions, profiles were calculated on a day-by-day basis for each night at which the application of the Raman lidar methods was possible, i.e., when low clouds and fog was absent. For the most favorable measurement period, the collected signal profiles were averaged, typically over 60-180 minutes, and corrected for background noise and system-dependent effects, such as the incomplete overlap between laser beam and receiver field-of-view in the lowermost 1.5 km above the lidar (Hofer et al., 2017). Raman lidar profiles of the particle extinction coefficient and extinction-to-backscatter ratio (lidar ratio) at 355 and 532 nm could be obtained for 276 nights.



The 532 nm particle backscatter coefficient and linear depolarization-ratio profiles are input in the POLIPHON (polarization lidar photometer networking) data analysis to derive height profiles of dust mass concentration, dust mass fraction, INP-relevant aerosol parameters, and of CCN and INP concentrations (Mamouri and Ansmann, 2016, 2017). The POLIPHON methodology could be applied to 328 nighttime observations. The technique was recently discussed with focus on desert dust

by Ansmann et al. (2019b). A case study from Dushanbe was shown in that article to provide an overview about the potential of the POLIPHON method. Thus, only a brief description is given here.

In a first step, dust and non-dust optical and associated microphysical properties are separated based on typical particle linear depolarization ratio values (Müller et al., 2007; Tesche et al., 2009) for dust (0.31) and non-dust (≤0.05). In Central Asia, the non-dust aerosol component covers contributions of anthropogenic haze and biomass burning smoke. The separated

backscatter profiles are converted into dust and aerosol-pollution extinction profiles by using typical lidar ratio values for Central Asian, Middle East, and eastern Saharan dust of (30–40 sr) and for Central Asian aerosol pollution (30–50 sr). The lidar-ratio observations will be presented in a follow-up article.

The dust and non-dust extinction profiles are then directly converted to number, volume and surface-area concentration profiles by means of the conversion factors listed in Table 1. In this article, we mainly concentrate on the retrieval of height profiles

of dust mass concentrations and cloud-relevant dust properties (CCN and INP concentration). The required dust conversion factors and parameters are derived from extended Aerosol Robotic Network (AERONET) observations at Dushanbe (Ansmann et al., 2019b). For the non-dust extinction-to-volume conversion, we used typical fine-mode conversion factors as presented by Mamouri and Ansmann (2017) for Central Europe (Leipzig, Germany). Similarly, we used Leipzig conversion parameters to obtain estimates for the non-dust CCN concentrations for Dushanbe, but assumed aerosol pollution background conditions,

i.e., a factor of 2 less fine-mode particles for a given non-dust extinction coefficient then in highly polluted Central Europe (Haarig et al., 2019a). Volume concentration profiles are converted to mass concentration profiles by using generic densities of dust (2.6 g cm$^{-3}$) and non-dust (1.5 g cm$^{-3}$) (Ansmann et al., 2012). To finally estimate INP concentrations for the most relevant ice nucleation modes (immersion freezing, deposition nucleation) (Mamouri and Ansmann, 2016), parameterizations are applied with dust particle number concentration $n_{250,d}$ (considering particles with radius larger than 250 nm only) as input

to obtain immersion-freezing INP concentrations (DeMott et al., 2015) and dust particle surface-area concentration $s_d$ as input to obtain deposition-nucleation INP concentrations (Ullrich et al., 2017). In the latter retrieval, the ice super saturation level during the ice crystal nucleation process is reuqired and a typical value of 1.15 (115% relative humidity over ice) is assumed. Dust CCN concentrations are estimated from profiles of number concentration of dust particles with a radius larger than 100 nm $n_{100,d}$ (Ansmann et al., 2019b; Lv et al., 2018). In the estimation of the aerosol-pollution-related CCN concentration, the dry

activation radius is assumed to be 50 nm. The respective conversion parameter $C_{n_{60,c}}$ in Tab. 1 with index 60 considers that particles at ambient aerosol conditions are usually slightly larger than dry particles. The conversion factor $C_{n_{60,c}}$ assumes that hygroscopic haze particles with radius >60 nm (at ambient conditions) are representing dry haze particles with dry radius of > 50 nm.





The Dushanbe lidar site was collocated with an AERONET sunphotometer station (AERONET, 2019; Holben et al., 1998) which is operated since 2010 (Abdullaev et al., 2012). The sun photometer provides AOT at 8 wavelengths and further particle optical and microphyical properties retrieved from the column-integrated daytime measurement (Hofer et al., 2017).

As auxilliary meteorological observations we used GDAS (Global Data Assimilation System) temperature and pressure profiles from the National Weather Service's National Centers for Environmental Prediction (NCEP) for the coordinates of 39° N and 69° E (GDAS modeling resolution of 1°) (GDAS, 2019). The temperature and pressure profiles are required in the lidar data analysis for the correction of air backscatter and extinction effects and also, e.g., in the computation of relative humidity from the water-vapor mixing ratio profile (Dai et al., 2018).

Furthermore, the HYSPLIT model (Hybrid Single Particle Lagrangian Integrated Trajectory Model) (HYSPLIT, 2019; Stein et al., 2015; Rolph, 2016) based on 1° GDAS reanalysis data was used to calculate backward trajectories. From March 2015 to August 2016 daily 120 h backward trajectories were calculated for Dushanbe arrival heights of 1.5, 2.5, 4.5 and 7.5 km (above ground level, a.g.l.). To describe the general air mass origin and the long-range transport features over Dushanbe, a seasonally resolved HYSLPIT cluster analysis was performed based on the backward trajectories from 2009–2018. The main results are shown in the next section.

## 3 Results

### 3.1 Aerosol layering and main aerosol transport features

Based on the 328 aerosol profiles we analyzed the annual cycle of aerosol layering in the lower, middle, and upper troposphere over Dushanbe. By visual inspection we found two main regimes: (a) the main aerosol layer that typically extends from the surface to about 3-6 km height and contributes to 500 nm AOT by usually more than 90%, and (b) frequently occuring thin dust layers between 5 and 10 km height that mainly contained aerosol from remote source regions such as the Arabian deserts and Saharan desert. We call the top height of the highest layer containing dust the uppermost aerosol layer top in the following discussion. A measurement example of this layering is shown in Fig. 2.

Besides the visual inspection of the lidar backscatter profiles, we tested several automated top-height detection methods. The most useful approaches (three in total) are considered in Fig. 2b and c. The first technique searches for the height (of the main layer) at which the backscatter coefficient (bsc) at 1064 nm wavelength drops below a threshold value of 2.5e-5 m$^{-1}$ sr$^{-1}$ (in Fig. 2b) for the first time above a starting height of, e.g., 500 m above the lidar. The second approach analyses the 1064 nm backscatter ratio (ratio of total-to-Rayleigh backscatter) profile and uses a threshold value (bsc ratio) of 1.8 (in Fig. 2c). The agreement between the different results is good.

In Fig. 2b, the blue dashed horizontal line shows the height level at which the integrated backscatter coefficient (IB, column backscatter) from the surface up to this specific height reaches 90% of the total column backscatter value IB (third method). The 90%IB height level of about 4 km means that most of the aerosol is in the main aerosol layer reaching to about 5 km on this day. The depolarization ratios for 355 and 532 nm in Fig. 2c, are far below the depolarization ratios for pure dust of 0.25 (355 nm) and 0.30 (532 nm) and thus indicate a mixture of mineral dust and air pollution.





Figure 3 provides an overview of the 18-month observations in terms of top heights of the main aerosol layer and the upper-most lofted aerosol layer, obtained from the visual inspection. In 184 cases out of the 328 available nighttime observations, a lofted aerosol layer (above the main layer) was present. As can be seen, the top height of the main layer increases from about 2.5-4.5 km in March to about 4-6.5 km in July 2015. From August 2015 the top heights decrease to minimum heights of about 1-2 km in December 2015 and January 2016. During February and March 2016 the top heights increased steeply. Later in the year 2016, the top heights are rather variable from day to day.

The top heights of the detected uppermost dust layer also vary strongly and indicate the frequent occurrence of dust traces up to the upper troposphere from late winter (February) to autumn (October). Such lofted layers are seldom during November to January. Although optically thin, dust layers in the middle and upper troposphere may have a sensitive impact on ice formation in mixed pahse and ice clouds (Ansmann et al., 2019a). Figure 4 provides a statistical overview of the top heights of the main and uppermost layer for the entire 18-month measurement period.

Table 2 summarizes the seasonal mean top heights of the main aerosol layer and the detected highest layer in the troposphere. The results obtained by case-by-case visual inspection and by applying the automated retrievals for the main layer depth are in good agreement (mostly within a range of 10% deviation). The seasonal means for the 90%IB level height indicate that most of the time the aerosol within the main layer (Fig. 2b) contributes 90% or even more to the overall IB or AOT (as will be shown later). As a consequence, we may conclude from the height-resolved observations that many snow-covered (glaciated) regions of Central Asia (Pamir mountains, Tien Shan) located at heights below 5 km (Treichler et al., 2018) are continuously exposed to dust and aerosol pollution during the summer half year.

Figure 5 provides an overview of the main air flow and aerosol transport towards Dushanbe. During the winter season (Fig. 5a), local sources and regional aerosols (clusters 1,2, and 5, 84%) contribute to the aerosol conditions in the main aerosol layer over Dushanbe. Source regions are northern Afghanistan, southwestern Tajikistan, Turkmenistan, southern Uzbekistan and the region downwind the Caspian Sea and the Aralkum Desert.

During the summer season, again regional air mass transport prevails (Fig. 5b, clusters 2, 3, and 4, 80%) in the main aerosol layer. Source regions are Uzbekistan, western Tajikistan, Aralkum Desert, and southern Kazakhstan. Regarding the upper tropospheric air mass transport Fig. 5c shows that the aerosol originates mainly (62%) from Middle East deserts (cluster 5) and North Africa (clusters 2 and 4), but also from polluted Mediterranean (cluster 1) and eastern European regions (cluster 3).

The cluster analysis suggests that the air masses are transported further to the east, crossing eastern Asia, continuously diluting, but mixing with new dust and pollution over China, and traveling across the Pacific. The upper tropospheric dust and aerosol pollution mixtures as observed over the lidar station at Dushanbe will become part of the northern hemispheric upper tropospheric aerosol background reservoir that influences cirrus formation and precipitation processes on continental to hemispheric scales.

## 3.2 Aerosol optical properties: AOT, DOT, and particle extinction profile

The characterization of the aerosol optical properties is based on the lidar and AERONET sun photometer observations. Figure 6 provides an overview of the basic optical properties obtained from the lidar measurements. For the comparison with




AERONET products (column-integrated values), the lidar-derived 532 nm AOT was determined from the particle extinction profile in Fig. 6b. By means of the Raman lidar method (Ansmann et al., 1992; Baars et al., 2012) the 355 and 532 nm particle extinction coefficients are directly computed from the respective 387 and 607 nm nitrogen Raman signal profiles for heights >1 km above the lidar. The large uncertainty in the correction of the incomplete laser-beam receiver field-of-view overlap pro-

hibits a trustworthy extinction coefficient retrieval from the nitrogen Raman signal profiles in the near range, i.e., for the lowest 1000 m above the lidar. To extend the particle extinction profiles towards the ground, we use the 532 nm particle backscatter coefficient in Fig. 6a. This quantity is obtained from ratio of the elastic backscatter signal to the respective nitrogen Raman backscatter signal (Ansmann et al., 1992; Baars et al., 2012) so that overlap effects widely cancel out. The 532 nm particle backscatter coefficients are trustworthy down to about 100 m above the lidar. We estimate the respective particle extinction co-

efficient for heights below 1-1.5 km by multiplying the backscatter coefficient with the lidar ratio of the actual measurement, in the present case measured at about 1.4 km height as shown in Fig. 6c and indicated by a dashed magenta line. The 532 nm AOT is finally obtained by the integration of the entire extinction profile up to 6 km height. In this way, we obtained 276 nighttime extinction profiles. Residual AOT contributions from heights above 6 km were usually <0.02.

To check the accuracy of lidar-derived AOT values we compared the lidar-derived AOTs with respective AERONET 500 nm

AOTs measured with the sun photometer in the afternoon, preferably close to sunset. For this purpose, 192 lidar extinction profiles were computed. The averaging time ranged from 15 min to 1 h 40 min with an average of about one hour. The temporal distance to the last AERONET measurement ranged from 2 hours to about 5 hours. Figure 7 shows the comparison. The agreement is acceptable. A small bias is observed and most probably related to the fact that the AERONET observations are performed at relatively low sun elevation angle across the city center of Dushanbe and lidar observations were performed in

the vertical direction. The combined AERONET and lidar field site was located in a less urbanized area, about 4 km east of the city center. The variability in the data are caused by temporally and spatially varying aerosol conditions and variations in the wavelength dependence of AOT in the 500-532 nm range and thus aerosol size distribution changes. The uncorrected wavelength dependence can cause differences of the order of 5%, only. It is interesting to note that a similar bias (deviation from the Dushanbe AERONET observations) was found when comparing MODIS (Moderate Resolution Imaging Spectroradiometer)

long-term observations of 550 nm AOT with the AERONET data shown in the supplementary material of Rupakheti et al. (2019). One of the reasons could be an overestimation of multiple scattering effects (caused by mineral dust) in the AERONET data analysis.

AOT histograms obtained from all 276 Dushanbe lidar nighttime profiles and all sun photometer AOT measurements (from March 2015 to Augsut 2016) (AERONET, 2019) are shown in Fig. 8. Similar AOT distributions are observed with both

instruments. Most AOTs are <0.3. Large values (>0.5) indicate dust outbreak events. The specific value of a polarization lidar is the potential to separate the dust from non-dust backscattering and thus to obtain an accurate estimate of the dust optical thickness (DOT). The distribution of DOT is given in Fig. 8c. Most DOT values are ≤0.2. This is in agreement with findings of Li and Sokolik (2018). These authors concluded from in-depth analysis of long-term spaceborne passive remote sensing over Central Asia that most DOTs are below 0.2 at 550 nm. Higher AOTs over western than over eastern Tajikistan were observed



and as a general finding, more dust was observed in the western parts of Central Asia (closer to the Caspian Sea) than in the eastern parts.

Rupakheti et al. (2019) analyzed satellite (MODIS) observations from 2002–2017 and found for the southern part of Central Asia (for the two southern countries Turkmenistan and Tajikistan, see Fig. 1) seasonal mean 500 nm AOT values of 0.18 (Tajik-
stan) to 0.22 (Turkmenistan) in spring, 0.2–0.22 in summer, 0.16-0.18 in autumn, and 0.14–0.17 during the winter months. In contrast, the Dushanbe AERONET observations revealed, seasonally mean values of 0.19 (spring), 0.36 (summer), 0.23 (autumn), and 0.15 (winter) for the CADEX period from March 2015 to August 2016. The lidar observations provided seasonal mean DOT/AOT ratios of 0.5 in spring, 0.8 in summer, 0.6 in autumn, and 0.1 in winter. Thus, a considerable part of the AOT is caused by anthropogenic fine-mode aerosol pollution. Our findings are in reasonable agreement with the MODIS observa-
tions of the Ångström exponent (describing the AOT wavelength dependence in the spectral range from 440 to 870 nm) with seasonally mean values of 1.0–1.2 for Turkmenistan and 1.3–1.4 for Tajikistan. From the Dushanbe 2015-2016 AERONET photometer observations we obtained however lower seasonally mean Ångström exponents, namely 0.7 (spring), 0.5 (summer), 0.8 (autumn), and 1.15 (winter). The dust and non-dust aerosol fractions of the total aerosol burden is further discussed in the next sections.

Figure 9 shows the different contributions of different height ranges to the measured DOT. For AOT (not shown) the features are similar. In Figs. 3 and 4, it was shown that the main aerosol layer reaches, on average, to 4-5 km height during the summer half year. The planetary boundary layer (convective mixing layer) typically covers the lowermost 1.5 km (red AOT contribution) and contributes by about 50% to AOT in spring and summer and to 80% during the winter months. The remaining part of the main aerosol layer (from about 1.5 to 4.5 km height) causes almost the entire residual AOT contribution. Only 10% of the AOT
is caused by particles in the middle and upper troposphere during spring.

The 18-month climatology for the 532 nm extinction coefficient is shown in Fig. 10. The figure is based on the 276 height profiles discussed above. Typical particle extinction values are 25-50 $\mathrm{Mm}^{-1}$ in spring and autumn, 50-100 $\mathrm{Mm}^{-1}$ in summer, and 50-150 $\mathrm{Mm}^{-1}$ during the winter season. According to the shown seasonal mean extinction profiles, the main aerosol layer reaches up to 5–5.5 km in spring, summer autumn and to about 2 km in winter. A moderate atmospheric variability in terms
of particle extinction is observed in spring, autumn and winter, but a strong variability is found during the summer season. This is the result of partly major dust storms with extreme particle extinction values of the order of 1500 $\mathrm{Mm}^{-1}$ and related horizontal visibilities of 2 km and less. Near-surface extinction values are highest in winter during the domestic heating period. The pollution is then trapped in the shallow aerosol layer with depth of 2 km only.

### 3.3 Aerosol microphysical and cloud-relevant properties: Profiles of mass, CCN and INP concentrations

The POLIPHON method permits the conversion of dust and non-dust extinction coefficients into height profiles of dust and non-dust mass, CCN, and INP concentration. The full retrieval procedure, starting from the basic data sets of 532 nm particle backscatter and linear depolarization ratio profiles is described in Sect. 2 and shown in Fig. 11. Recently, the required conversion parameters factors for mineral dust were updated and include now Central Asian and Middle East dust conditions (Ansmann et al., 2019b). The measured particle depolarization ratio at 532 nm was close to 0.3 at heights above 3 km and





indicated the presence of an almost pure dust layer up to 8 km height. Only in the lower part (below 2.5 km height) the depolarization ratio dropped below 0.2 and indicates a mixture of mineral dust, aerosol pollution (urban industrial particles, biomass burning smoke), and continental background aerosol. This case was already discussed by Hofer et al. (2017).

In the shown example, the dust mass concentration was low ($<25$ $\mu$g m$^{-3}$) in the polluted layer and $>150$ $\mu$g m$^{-3}$ in the

center of the lofted dust layer. The mass concentration of continental aerosol pollution was much lower with values $<5$ $\mu$g m$^{-3}$ throughout the troposphere. The estimated profiles for the dust and non-dust CCN concentrations show values of up 300 cm$^{-3}$ in the center of the lofted dust plume and a total CCN concentration of about 150 cm$^{-3}$ in the polluted layer below 2 km height (see Fig. 11c). By means of the derived height profiles of dust particle number concentration considering particles with radius $>250$ nm only and the dust particle surface area concentration in Fig. 11 together with the respective GDAS temperature profile,

the profile segments for the INP concentrations are obtained in Fig. 11e. We distinguish profiles relevant for immersion freezing (DeMott et al., 2015) and deposition nucleation of ice crystals (Ullrich et al., 2017). Immersion freezing dominates in mixed-phase clouds at temperatures $> -30°$C, whereas deposition nucleation is the relevant heterogeneous ice nucleation process at temperatures $< -30°$C, e.g., in cirrus layers. We only show the dust INP values because non-dust aerosol components (such as soot particles) are inefficient INPs at the given temperatures.

Figures 12, 13, and 14 summarize the findings regarding dust and non-dust mass concentrations and dust mass fraction for the 18-month lidar campaign. According to Fig. 12, typical (mean) dust mass concentrations in the lowermost 2.5 km of the atmosphere 100 $\mu$g m$^{-3}$ (spring), 200-600 $\mu$g m$^{-3}$ (summer), 100-300 $\mu$g m$^{-3}$ (autumn), and 20-50 $\mu$g m$^{-3}$ (winter). The season mean values for anthropogenic haze and biomass burning smoke are an order of magnitude lower with typical values (up to 2.5 km height) of 10-20 $\mu$g m$^{-3}$ (spring to autumn), and 20-50 $\mu$g m$^{-3}$ during the domestic heating period.

Figure 13 shows the mean dust mass fraction as a function of height. Each mean profile is based on a different set of single profiles. In each of the 8 computations, we considered only profiles with significant particle backscatter and depolarization ratio up to the given top height from 3 km (in a) to 10 km (in h). This means, we did not consider measurements when the backscatter coefficient indicated clear air in the upper part of the profile, i.e., below the defined top height. The aim was to determine the mean dust fraction in the occurring dust layers. By excluding all cases with clear air in the upper part of the

averaging height range, the number of considered night-by-night observations decreases from 310 profiles (out of a total set of 328 profiles) in the case of the top height of 3 km (Fig. 13a) to 9 profiles when considering and averaging all profiles that show dust from the surface up to 10 km height (Fig. 13h). As can be seen, the profile-mean dust mass fraction is in most cases 70–80%. The dust mass fraction typically decreases with height from 90% to about 50% in the uppermost part of the profile. The other way around, the anthropogenic aerosol mass fraction is always of the order of 20-30% in the main aerosol layer up

to 4-5 km height, and also in the lofted dust layers higher up (5-10 km height range). This is in consistency with the DOT and AOT mean values and DOT/AOT ratios discussed above.

Figure 14 highlights the decreasing number of dust cases (available for dust profile averaging up to a given top height) with increasing height. Figure 14 is similar to Fig. 13, but considers a higher resolution concerning the defined top heights in the averaging procedure. The main result is a very smooth, decreasing curve for the frequency of occurrence of dust layers. In

terms of the number of observed dust cases, close to 100% (with 80% mean dust mass fraction) out of all 328 nighttime profiles





show dust up to 3 km top height, as already mentioned above, 65% in the case of the 5 km top height (mean dust fraction of 74%), and only a few cases of dust (<10%) when the defined top height is >9 km.

Figures 15 to 18 summarize the results concerning the cloud-relevant dust and non-dust aerosol parameters. As a general impression, CCN and INP concentrations decrease with height because of the decreasing dust particle number concentration. To facilitate the comparison of CCN and INP concentration levels for the different seasons we define typical height ranges as representative reservoirs for CCN and INPs. The 3-4 km height range can be regarded as the main CCN and immersion-freezing INP reservoir for convective cloud formation (liquid-water and mixed-phase clouds) over Dushanbe during the spring, summer, and autumn seasons. The INP levels at 7-8 km height may be representative for deposition INP reservoir in the upper troposphere and thus relevant for cirrus formation. Table 3 summarizes the layer mean values of the cloud-relevant aerosol quantities for the 3-4 km and 7-8 km height ranges and for the different seasons.

The results for CCN concentrations in Fig. 15 show that dust particles lead to mean dust CCN concentrations of the order of $50\,\mathrm{cm}^{-3}$ (spring), $100\,\mathrm{cm}^{-3}$ (summer), and 30-50 $\mathrm{cm}^{-3}$ (autumn) and non-dust CCN concentrations of the order of $200\,\mathrm{cm}^{-3}$ (spring), $400\,\mathrm{cm}^{-3}$ (summer), and 250 $\mathrm{cm}^{-3}$ (autumn) in the 3-4 km height layer during the summer half year (spring to autumn). The shown SD values (difference between the solid and dotted lines in Fig. 15) indicate an atmospheric variability according to a factor of two (thus a value range from almost zero to $2\times$ mean). For comparison, Schmale et al. (2018) summarized European CCN observations (for a supersaturation values of 0.2%) and compared them with other measurements in Alaska and Amazonia. They found mean CCN concentrations of 1200-1500 $\mathrm{cm}^{-3}$ in central Europe, 100 $\mathrm{cm}^{-3}$ at the Atlantic coast in Ireland, <100 $\mathrm{cm}^{-3}$ in Alaska during the summer half year, and 100 $\mathrm{cm}^{-3}$ in the wet season to 700 $\mathrm{cm}^{-3}$ in the dry season in Amazon, Brazil. Haarig et al. (2019b) reported CCN concentrations of 140-270 $\mathrm{cm}^{-3}$ in the lofted Saharan air layer (containing pure dust) over Barbados in the Caribbean during the summer season of 2013 and 400-500 $\mathrm{cm}^{-3}$ in spring (March 2014) for a mixture of dust and pollution over Barbados originating from Africa. Thus, the CCN levels over Dushanbe indicate moderately polluted continental aerosol background conditions.

Figure 16 provides an overview of the observations of dust properties that are used as aerosol input in the INP concentration retrievals. Dust is the main INP relevant aerosol component. The uncertainty in these input profiles is only 30%. The large uncertainty in the INP concentration estimates of a factor of 3-5 is caused by the use of the published INP parameterization schemes. Figure 17 shows seasonal mean INP concentration profiles for an air temperature of $-25°\mathrm{C}$ relevant for immersion freezing and for $-50°\mathrm{C}$ and thus a temperature relevant for deposition nucleation of ice crystals.

The seasonal mean values in the 3–4 km aerosol layer range from 0.9–3.3 $\mathrm{cm}^{-3}$ ($n_{250,\mathrm{d}}$), 20–75 $\mu\mathrm{m}^2\,\mathrm{cm}^{-3}$ (dust particle surface area concentration), and 2.3–12 $\mathrm{L}^{-1}$ (immersion freezing INP concentration), during the three summer seasons (spring to autumn). The atmospheric variability is again roughly described by a factor of 2. In case of cloud formation and the evolution of the ice phase, significant lifting of cloud parcels and cooling of the air parcels occur. Note that the INP concentration then increases by an order of magnitude for the given aerosol concentration when the air temperature decreases by 5 K triggered for example by the strong lifting in updrafts.

To compare the Dushanbe INP concentration levels with the ones in other dust regions we checked the literature. Price et al. (2018) performed airborne in situ measurements and found INP concentrations (immersion freezing, $-25°\mathrm{C}$) from 10–



1000 L$^{-1}$ in dust layers from 300 to 3500 m height over Cabo Verde in the summer of 2015 in the outflow regime of Saharan dust, less than 1000 km west of the African coast. The maximum values of the large particle number concentration (coarse mode fraction) reached 200 cm$^{-3}$ and the maximum particle surface area concentrations were as high as 1500 $\mu$m$^2$ cm$^{-3}$. DeMott et al. (2015) reported in situ measured values of $n_{250,\mathrm{d}} = 15 - 20$ cm$^{-3}$ and INP concentrations (immersion freezing, $-25°$C) of 60-100 L$^{-1}$ in a lofted dust layer around 2 km height over Cabo Verde in July 2011. Haarig et al. (2019b) discussed observations taken in the Saharan dust layer between 2-4 km height in the Caribbean in the summer of 2013, more than 5000 km west of the African dust sources km, and found INP concentrations of 30–60 L$^{-1}$ (immersion freezing, $-25°$C) and 10 L$^{-1}$ in the spring season (March 2014, 2-3 km height). Ansmann et al. (2019b) reported a mixed-phase cloud layer (at 5–6 km height) occurring in desert dust over Cyprus in the Eastern Mediterranean in March 2015, and the INP concentration was estimated to be of the order of 1 L$^{-1}$ at $-20°$C which means about 10 L$^{-1}$ at $-25°$C. Thus the seasonal mean Dushanbe INP concentration levels in 3-4 km height indicate typical INP conditions in regions influenced by long-range transport of dust.

The INP reservoir between 7–8 km height contains significant amounts of INPs over Dushanbe only during the spring and summer seasons. The atmospheric variability is high and characterized by a factor of 4–6 (one positive SD). The seasonal mean particle surface area concentrations ranged from 0.7–3.3 $\mu$m$^2$ cm$^{-3}$ in spring and summer and were around 0.057 $\mu$m$^2$ cm$^{-3}$ in autumn and indicate seasonal mean INP concentrations (deposition nucleation, $-50°$C, ice supersaturation of 1.15) of 2.4 L$^{-1}$ (spring), 0.55 L$^{-1}$ (summer), and 0.041 L$^{-1}$ in autumn. For comparison, Ansmann et al. (2019a) found over Cyprus in lofted dust at cirrus level (10-11 km height, $-50°$C) INP concentrations (deposition nucleation, for an ice supersaturation value of 1.1) of 5 L$^{-1}$ and corresponding particle surface area concentrations around 30 $\mu$m$^2$ cm$^{-3}$. The dust had a strong influence on the cirrus evolution and life cycle. Thus, during times with strong dust advection to Dushanbe at greater heights, predominantly during spring and summer, a significant impact of dust on ice formation in the upper troposphere can be expected.

As a final result we present the INP concentration profiles for the found aerosol conditions in combination with the actually observed pressure and temperature profiles in Fig. 18 to give an impression of typical (actual) INP concentration values for an ice supersaturation of 1.15. It can be seen that only in spring and summer significant levels of INP concentrations (0.01–1 L$^{-1}$) occurred in the 6-8 km height range (mixed-phase cloud and immersion freezing regime) and from 8-10 km height (ice cloud and deposition nucleation regime). In autumn and winter the seasonal means indicate a rather low potential for heterogeneous ice formation at the given height levels.

## 4 Conclusion/Outlook

Deteriorating environmental conditions expressed by melting glaciers, desiccating lakes and strong risks for further severe changes in near future, and on the other hand side the lack of advanced aerosol observations in Central Asia was the motivation for the 18-month CADEX campaign. The main results were presented here. For the first time, vertical profiling of the annual cycle of aerosol conditions over Dushanbe, Tajikistan with a state-of-the-art multiwavelength aerosol lidar was conducted. By applying modern data analysis techniques the mixtures of mineral dust and anthropogenic aerosol pollution were described in terms of DOT, AOT, seasonal mean height profiles of 532 nm particle extinction coefficient, dust and non-dust


mass concentration and dust fraction profiles, as well as in terms of cloud-relevant aerosol properties such as large particle number concentration $n_{250}$, particle surface area concentration, CCN and INP concentrations. These latter parameters describe the impact of aerosols on cloud formation processes. The Dushanbe lidar long term study demonstrates the strong potential of modern lidar instruments to contribute to aerosol and aerosol-cloud interaction research, and environmental (air quality) monitoring.

The key results can be summarized as follows. The main aerosol layer over Dushanbe (which may be a representative site for Central Asia) reaches typically 4-5 km height in spring to autumn so that most of the local glacier regions are exposed to polluted and dusty air throughout the year, except the winter period. Frequently lofted dust-containing aerosol layers were observed at heights from 5-10 km, indicating a sensitive potential of dust to influence cloud ice formation. Typical dust mass fractions were of the order of 60–80%, i.e., a considerable part of the aerosol is anthropogenic pollution and biomass burning smoke. The highest aerosol pollution levels over Dushanbe occur during the winter months. The seasonal mean 500 nm AOT ranges from 0.15 in winter to 0.36 in summer during the CADEX period (March 2015 to August 2016), DOTs were typically below 0.2, seasonal mean particle extinction coefficients were of the order of 100–500 Mm$^{-1}$ in the main aerosol layer during the summer half year, and about 100-150 Mm$^{-1}$ in winter, but mainly caused by anthropogenic haze.

Similarly, the highest dust mass concentrations occur in the summer season (200-600 $\mu$g m$^{-3}$) and the lowest during the winter months (20-50 $\mu$g m$^{-3}$) in the main aerosol layer. In winter, the anthropogenic aerosol pollution caused mass concentrations of 20-50 $\mu$g m$^{-3}$, while during the summer half year (spring to autumn) the mass concentration caused by urban haze and biomass burning smoke decreases to 10-20 $\mu$g m$^{-3}$ in the lowest part of the troposphere. The derived CCN concentration levels indicate moderately polluted aerosol background conditions. The INP concentrations during spring and summer seasons were found to be high enough in the middle and upper troposphere to significantly influence ice formation in mixed-phase and ice clouds. During autumn and winter, however, the INP concentration levels are very low in the 5-10 km height range.

As an outlook, there is clear request for continuous monitoring and documentation of environmental conditions (and changes during the upcoming years) in Central Asia by means of in situ measurements (surface network) and ground-based and spaceborne active and passive remote sensing. The Central Asian region must be better integrated into international research activities. It is a key region of climate change, air pollution, degradation of living conditions. Continuous network observations are needed to support decision makers, the atmospheric research community, and weather/dust prediction and environmental services. From the point of view of lidar profiling of aerosols, we need an extension of the existing well organized ground-based lidar network infrastructure in Europe and eastern Asia to cover this key region of climate and environmental changes which ranges from the Eastern Mediterranean over the Middle East to western China. Coherent aerosol profile observations are required to improve our understanding of northern hemispheric dust and aerosol pollution long-range transport and aerosol life cycles as a whole. As a first step to improve the current unsatisfactory network situation in Central Asia, we deployed a new Polly lidar at Dushanbe in June 2019 to conduct continuous observations over the next 5-10 years. Besides tropospheric monitoring, we also need more ground-based systems in this region of the world for an improved stratospheric aerosol monitoring. Complex mixtures of volcanic ash, sulfuric acid droplets and soot particles occur and need to be documented and characterized in detail by modern lidar networks to assist atmospheric modeling and climate prediction efforts.



*Acknowledgements.*  The CADEX project was funded by the German Federal Ministry of Education and Research (BMBF) in the context of "Partnerships for sustainable problem solving in emerging and developing countries" under the grant number 01DK14014. The construction of a new lidar for permanent observations in Tajikistan is funded by the BMBF under the grant number 01LK1603A. This project has also received funding from the European Union's Horizon 2020 research and innovation program ACTRIS-2 Integrating Activities (H2020-INFRAIA-2014-2015, grant agreement no. 654109) and from the European FP7 project by the European Union's Seventh Framework Program (FP7/2007-2013) collaborative project BACCHUS (grant agreement no. 603445).





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





**Table 1.** Applied values of the conversion parameters required in the POLIPHON retrieval (Mamouri and Ansmann, 2016, 2017; Ansmann et al., 2019b). The conversion factors are explained in the text and are needed to convert particle extinction coefficients into particle mass concentrations and cloud-relevant parameters (CCN and INP concentrations). Index d and c denote dust and continental fine-mode aerosol pollution, respectively.

| Parameter | Value | Uncertainty | Unit |
|---|---|---|---|
| $C_{v_d}$ | 0.79e-12 | 0.1e-12 | Mm |
| $C_{s_d}$ | 3.11e-12 | 0.6e-12 | Mm m$^2$ cm$^{-3}$ |
| $C_{n_{250,d}}$ | 0.135 | 0.0278 | Mm cm$^{-3}$ |
| $C_{n_{100,d}}$ | 12.4 | 3 | Mm cm$^{-3}$ |
| $b_d$ | 0.71 | 0.05 | - |
| $C_{n_{60,c}}$ | 25.7 | 1.7 | Mm cm$^{-3}$ |
| $b_c$ | 0.94 | 0.03 | - |
| $C_{v_c}$ | 0.24 | 0.08 | Mm |

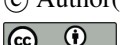



**Table 2.** Mean value and standard deviation of the top height of the main aerosol layer and of the uppermost detected layer (by individual inspection, no automated retrieval), and by automated retrieval (two different methods with threshold values of bsc=2.5e-5 m$^{-1}$ sr$^{-1}$, bsc ratio of 1.8) in the case of the main layer top height. Mean and SD values are given separately for spring (MAM), summer (JJA), autumn (SON), and winter (DJF). The numbers of the considered nighttime observations per season are given in brackets. IB denotes the integrated (column) backscatter coefficient, and the given height at which the 90%IB level is reached indicates roughly the vertical extend of the layer that contributes to AOT by 90%. For more explanations of the automated retrieval methods (two last lines) and of the abbreviations IB, bsc, and bsc ratio see text.

| Season | MAM (88) | JJA (144) | SON (59) | DJF15/16 (37) |
|---|---|---|---|---|
| Main layer depth (km) | 4.3±1.2 | 4.7±0.9 | 3.6±0.9 | 2.9±1.2 |
| Uppermost layer top height (km) | 7.7±1.8 | 7.0±1.3 | 6.4±1.8 | 5.5±1.8 |
| 90%IB height (km) | 4.6±1.4 | 3.7±0.7 | 2.8±1.8 | 3.0±1.2 |
| bsc = 2.5e-5 m$^{-1}$ sr$^{-1}$ threshold height (km) | 5.5±1.7 | 5.2±1.2 | 3.7±1.0 | 2.9±1.1 |
| bsc ratio = 1.8 threshold height (km) | 4.6±1.7 | 5.0±1.1 | 3.4±1.1 | 2.3±1.3 |





**Table 3.** Overview of layer mean values of INP-relevant aerosol properties (n250 for large particle concentration, sa for surface area concentration) and CCN and INP concentrations for the 3–4 km and 7–8 km layers for spring (MAM), summer (JJA), autumn (SON), and winter (DJF). Immersion-freezing INP concentrations (INP-imm) are given for $-25°C$ and deposition-nucleation INP concentrations (INP-dep) for $-50°C$ and an ice supersaturation value of 1.15. The range of values is indicated by the sum of the mean values $+SD$.

| Height range [km] | 3-4 | 7-8 | 3-4 | 7-8 | 3-4 | 7-8 | 3-4 |
|---|---|---|---|---|---|---|---|
| Season | MAM | MAM | JJA | JJA | SON | SON | DJF |
| Dust CCN (CCN+SD) [cm$^{-3}$] | 52 (117) | 9 (25) | 108 (180) | 2 (10) | 34 (85) | 0.3 (1.6) | 7 (27) |
| Non-dust CCN (CCN+SD) [cm$^{-3}$] | 214 (401) | 32 (81) | 425 (650) | 13 (38) | 250 (511) | 5 (23) | 104 (228) |
| Dust n250 (n250+SD) [cm$^{-3}$] | 1.4 (3.9) | 0.14 (0.53) | 3.3 (6.4) | 0.03 (0.16) | 0.9 (2.5) | 0.002 (0.015) | 0.15 (0.63) |
| Dust sa (sa+SD)[10$^{-12}$ m$^2$ cm$^{-3}$] | 32 (89) | 3.3 (12.2) | 75 (147) | 0.76 (3.8) | 20 (58) | 0.06 (0.36) | 3.4 (14.5) |
| Dust INP-imm (INP+SD) [L$^{-1}$] | 4 (8) | – | 12 (24) | – | 2.3 (4.7) | – | 0.24 (0.49) |
| Dust INP-dep (INP+SD) [L$^{-1}$] | – | 2.4 (8.9) | – | 0.6 (2.7) | – | 0.04 (0.26) | – |



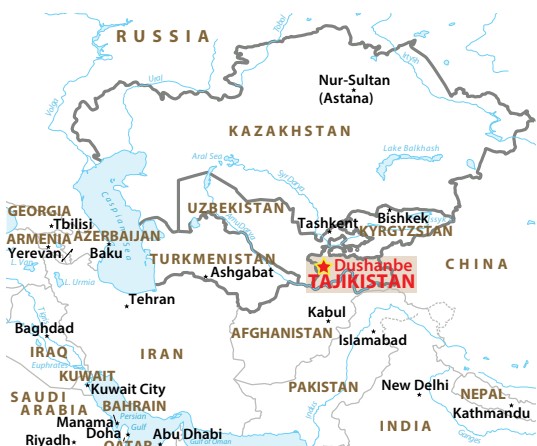

**Figure 1.** The five countries defining Central Asia (within the thick boundaries). Highlighted is the lidar station (red star) at Dushanbe, Tajikistan (http://www.shadedrelief.com/political/Political_Map_Pat.pdf, adapted).

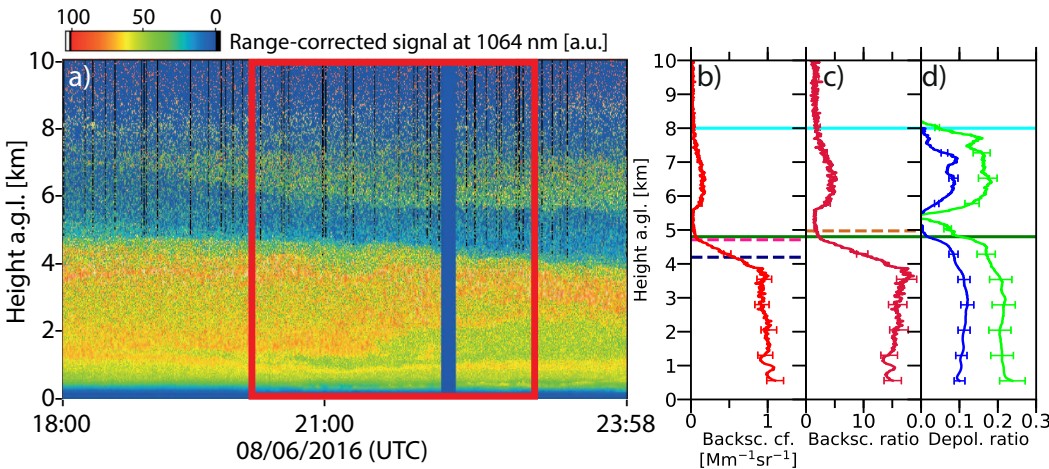

**Figure 2.** a) Aerosol layering over Dushanbe, Tajikistan, observed with lidar on 8 June 2016 from 18:00-23:58 UTC in terms of the range-corrected signal at 1064 nm wavelength. The vertical blue column indicates the time interval of automated depolarization calibration (excluded for the data analysis in b to d). From the signal profiles collected within the period from 20:00 to 22:59 UTC (red box in a), mean profiles of (b) the particle backscatter coefficient at 1064 nm (23 m vertical smoothing length), (c) respective 1064 nm backscatter ratio, and (d) particle linear depolarization ratio at 355 nm (blue) and 532 nm (green) are calculated (308 m vertical smoothing length). Error bars indicate the uncertainty in the computed values. The solid green and light blue horizontal lines in (b)-(d) indicate the manually determined top heights of the main aerosol layer and of the uppermost aerosol layer, respectively. The red dashed lines in (b) and (c) show the respective top heighs when using the backscatter threshold methods with the threshold values of bsc=2.5e-5 m$^{-1}$ sr$^{-1}$ (in b) and bsc ratio of 1.8 (in c). The blue dashed line in (b) indicates the height $z$ at which column integrated backscatter IB($z$) value reached the 90%IB level.

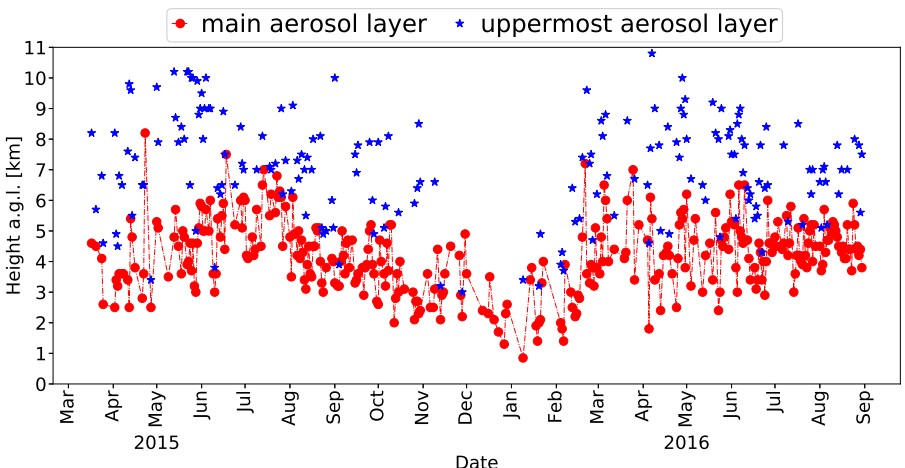

**Figure 3.** Top heights of the main aerosol layer (red) and of the uppermost aerosol layer (blue). All lidar backscatter profiles were inspected manually, i.e., an automated retrieval of layer top heights was not applied.

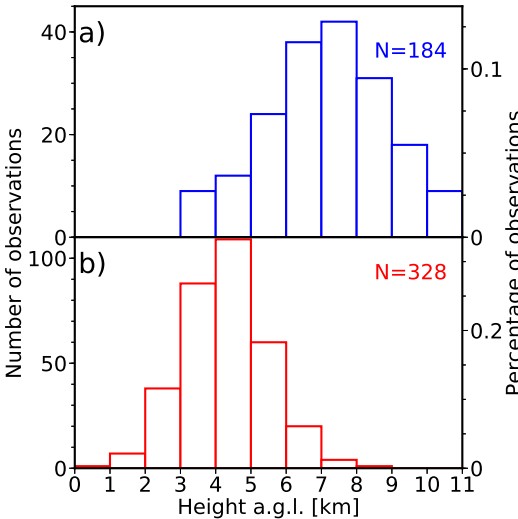

**Figure 4.** Histograms of layer top heights for (a) the uppermost layer (blue) and (b) the main aerosol layer (red). Total number of observations is 328.



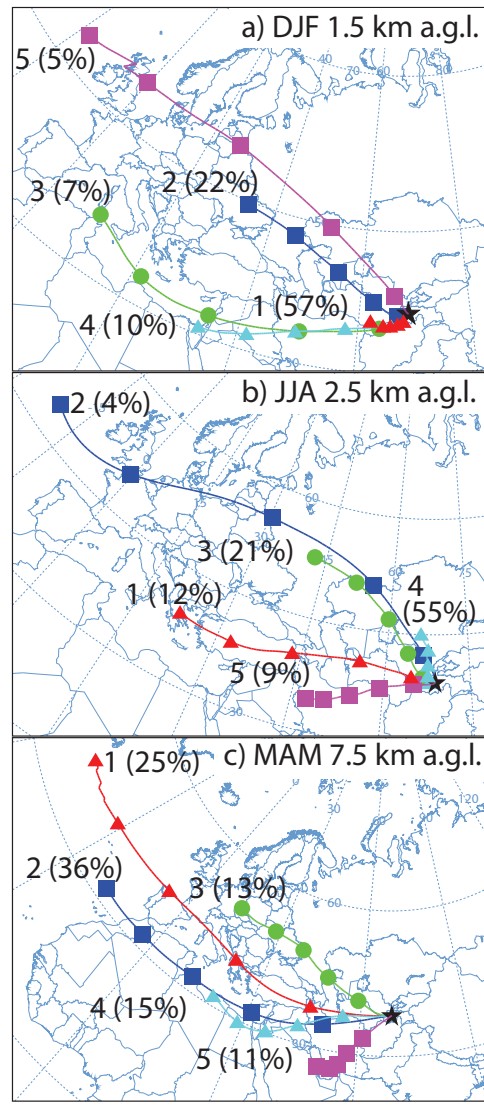

**Figure 5.** Results of the seasonally resolved HYSPLIT cluster analysis of daily 120 h backward trajectories arriving above Dushanbe at (a) 1.5 km height a.g.l. (winter season, DJF), (b) 2.5 km height (summer season, JJA), and (c) 7.5 km height (spring season, MAM). 10 years (2009–2018) of daily HYSPLIT backward trajectories are considered, about 900 trajectories per season. Five clusters are determined for each season. The relative frequency of occurrence of air mass transport belonging to a specific cluster is given in percent at the beginning of the cluster trajectory together with the cluster number. Local and regional aerosol sources control the environmental conditions in the main aerosol layer (below 4-5 km height) in winter (a) as well as in summer (b), while long-range transport of dust from the Middle East deserts and the Sahara mainly determine the aerosol conditions in the upper troposphere (above the main aerosol layer) in spring (c) and summer (not shown).

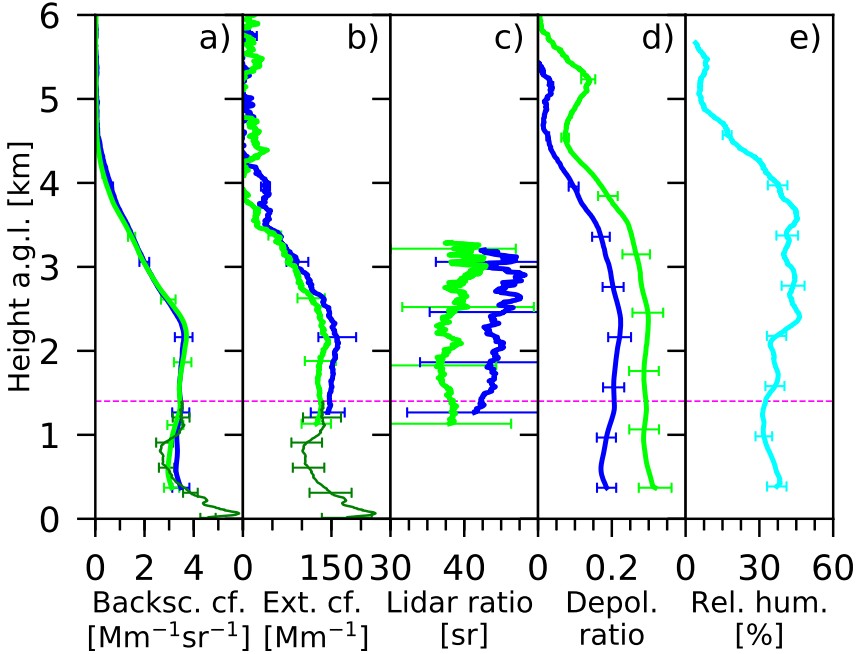

**Figure 6.** Overview of the basic lidar products obtained from the Dushanbe lidar observations. The Polly measurement was taken on 28 June 2016, 18:00-20:59 UTC: (a) Particle backscatter coefficient at 355 nm (blue, 743 m vertical smoothing length) and 532 nm (dark green, 23 m vertical smoothing, green, 743 m vertical smoothing), (b) particle extinction coefficient at 355 (blue) and 532 nm (green), 743 m vertical smoothing, and 532 nm particle extinction coefficient (dark green) calculated from the 532 nm particle backscatter coefficient (dark green in b) and the 532 nm lidar ratio at 1.4 km (magenta dashed line indicates the 1.4 km height), (c) lidar ratio at 355 (blue) and 532 nm (green), 743 m vertical smoothing, (d) particle linear depolarisation ratio at 355 and 532 nm, and (e) relative humidity. The 532 nm is computed from the extinction profiles segments from the surface to 1.4 km height (in b, dark green) and from 1.4–6.0 km (in b, green). Error bars show the uncertainty in the profile data.





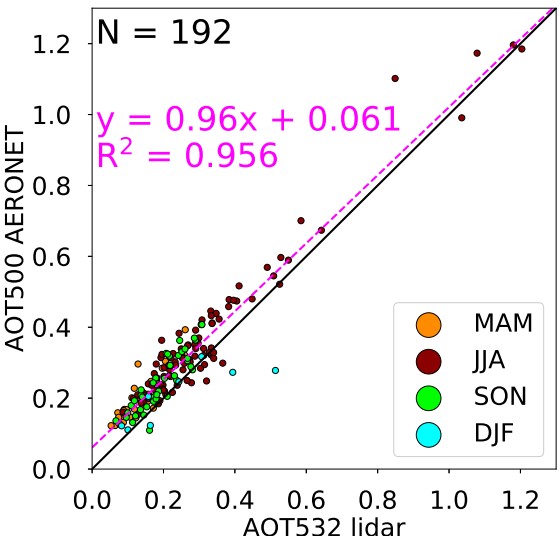

**Figure 7.** Lidar-derived 532 nm AOT vs AERONET 500 nm AOT. N=192 lidar observations (performed after sunset, outside the city center, vertical pointing) are compared with AERONET measurements before sunset (frequently at low sun elevation angle, measurement path across polluted Dushanbe). Different seasons are contrasted by different colors (spring, MAM, summer, JJA, autumn, SON, winter, DJF). The dashed magenta line is obtained from linear regression analysis, corresponding equation and correlation coefficient are given in magenta as well.

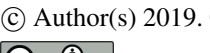



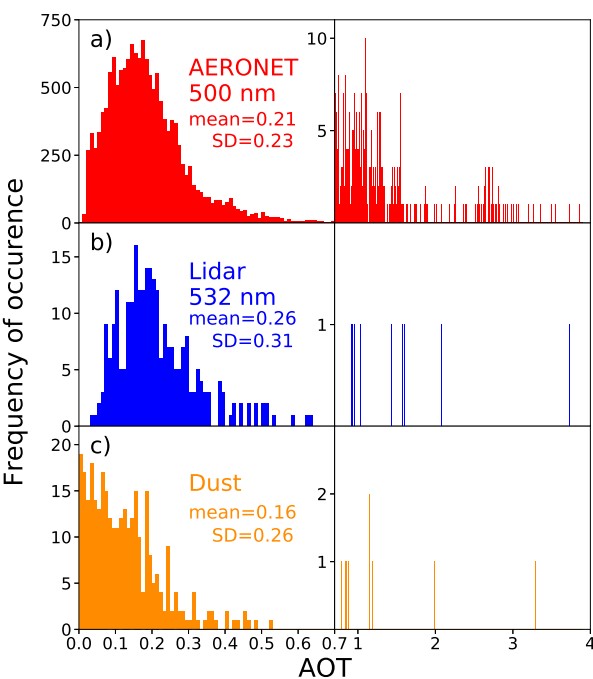

**Figure 8.** Histograms of (a) AERONET 500 nm AOT, (b) lidar-derived 532 nm AOT, and (c) lidar-derived dust optical thickness (DOT). DOT is obtained from the height profile of the dust-related backscatter coefficient multiplied by a typical dust lidar ratio of 35 sr.

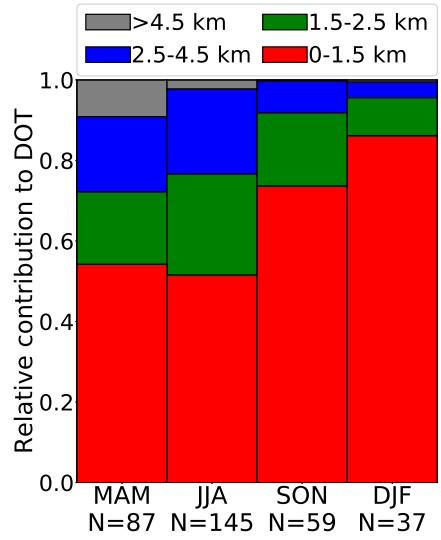

**Figure 9.** Seasonally-resolved relative contributions of different height regions to DOT (532 nm). The analysis is based on the data shown in Fig. 8c and the respective height profiles of the particle extinction coefficient. Numbers of available observations are given in the lowest line. The figure is similar for AOT (not shown).





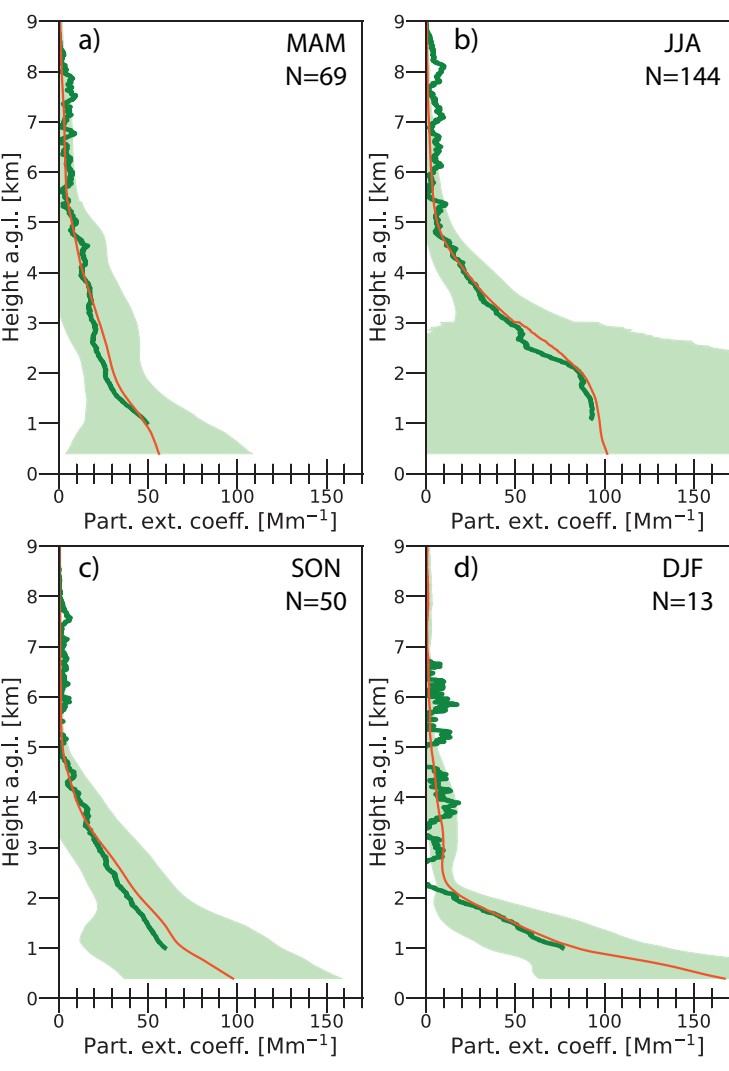

**Figure 10.** Seasonal mean 532 nm total (dust + non-dust) particle extinction coefficient for (a) spring, (b) summer, (c) autumn, and (d) winter. The thick dark green line shows the mean extinction coefficient directly computed from the nitrogen Raman signal profiles. The red line is obtained from the same lidar nightime observations but in terms of the respective 532 nm backscatter coefficient profiles multiplied by a lidar ratio of 35 sr (in a-c, spring, summer, autumn) and 50 sr (in d, winter). The green shadow shows the atmospheric variability (one standard deviation) based on the backscatter coefficient profiles per season. The N values show the number of available Raman lidar observations per season. The maximum near-surface aerosol extinction coefficient was about 1500 $Mm^{-1}$ occurring in the summer season during severe dust outbreaks.



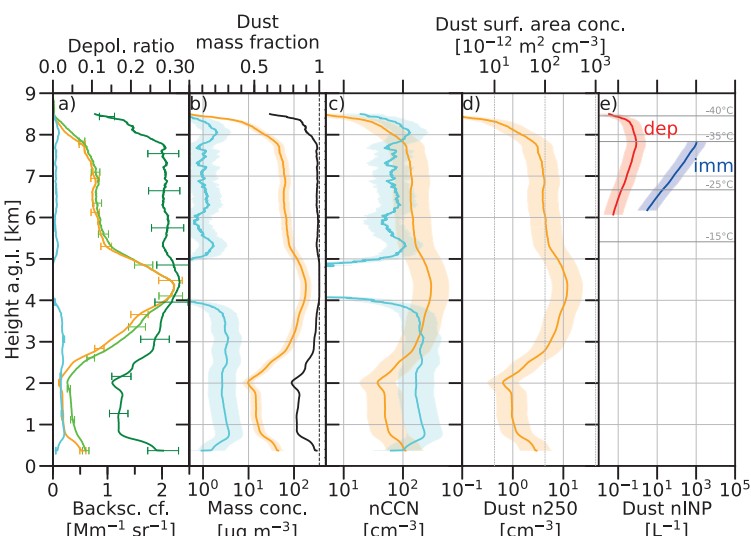

**Figure 11.** Overview of microphysical and cloud-relevant particle properties obtained by applying the POLIPHON method to the polarization lidar measurements. The lidar measurement was performed on 23 April 2015, 21:00-21:34 UTC. (a) The 532 nm particle backscatter coefficient (in green) and the particle linear depolarisation ratio (in dark green) are used as input to obtain the results in (a)-(e). The POLIPHON products are (a) the derived 532 nm dust backscatter coefficient (yellow) and the non-dust backscatter coefficient (light blue), (b) dust mass concentration (yellow) and the non-dust mass concentration (light blue), and the dust mass fraction (black, ratio of the dust to total particle mass concentration, dashed black vertical line shows a dust mass fraction of 1), (c) dust and non-dust CCN concentrations (nCCN), (d) dust particle number concentration $n_{250}$ and surface area concentration, and (e) dust-related INP concentration profiles when applying the INP parameterizations for deposition nucleation (dep, in red, use of the dust surface area concentration as input) and immersion freezing (imm, in blue, with $n_{250}$ as input). Horizontal lines in (e) show the temperature levels on 23 April 2015. Error bars and ranges (shadows) indicate the uncertainty in the retrieved values.

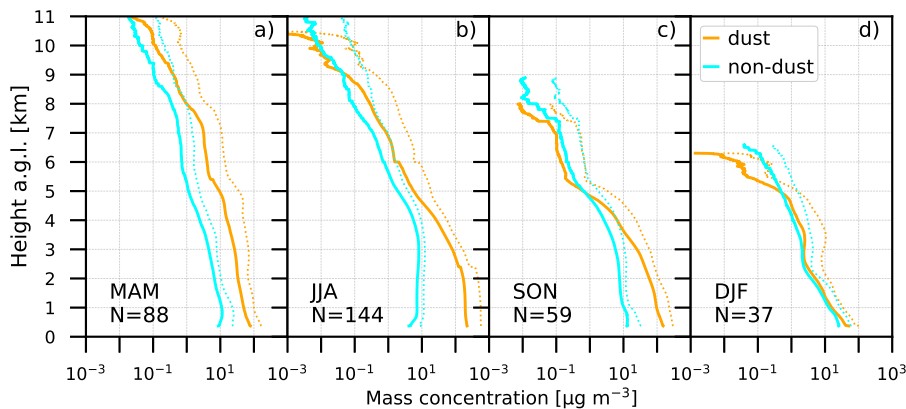

**Figure 12.** Seasonal mean dust (yellow) and non-dust (cyan) mass concentration profiles for (a) spring, (b) summer, (c) autumn, and (d) winter. The dotted lines show the (mean + SD) values and provide an impression of the atmospheric variability. The (mean − SD) values are close to zero. Considered nighttime observations N are given as numbers.

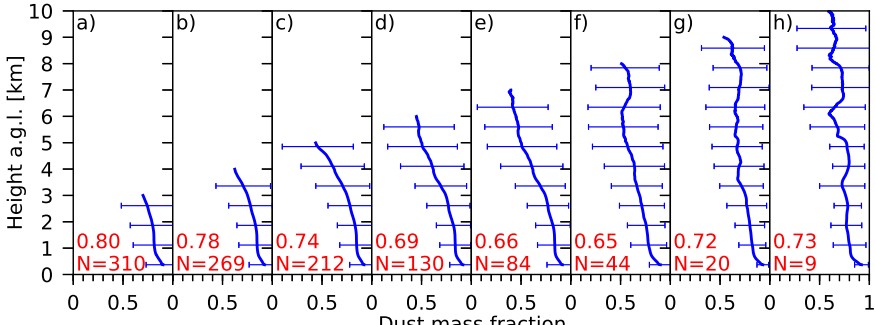

**Figure 13.** Mean dust mass fraction as a function of height up to (a) 3 km, (b) 4 km, (c) 5 km, (d) 6 km, (e) 7 km, (f) 8 km, (g) 9 km, and (h) 10 km. Each profile considers a different (decreasing) number N of observations (given in each of the eight panels, total number of observations is 328). Only observations with a backscatter coefficient clearly above zero up to the top height of the profile so that a dust mass fraction could be calculated are considered. Profiles showing pure Rayleigh scattering in the upper part of the height profiles (below the defined (a) - (h) top height) are excluded from the averaging. The bars indicate the atmospheric variability (1 SD). The profile mean dust mass fraction obtained from the shown 8 mean dust profiles are given as numbers together with the available number N of profiles in each of the 8 averaging processes (see text for more details).





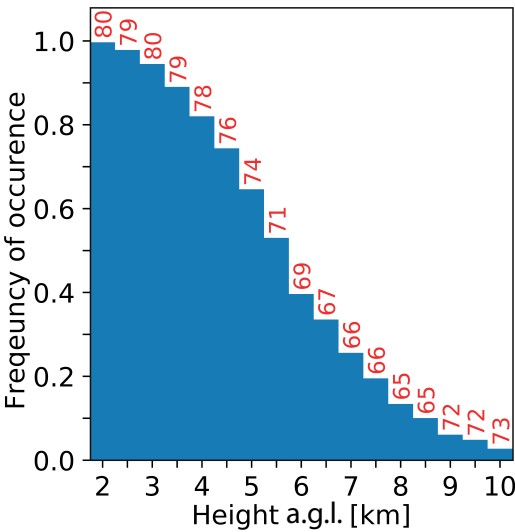

**Figure 14.** Relative frequency of occurrence of dust up to a given top height, ranging from 2-10 km top height (shown with a top height resolution of 0.5 km). The given red numbers are column mean dust mass fractions calculated in the same way as in Fig. 13. Total number of available dust mass fraction profiles is again N=328.

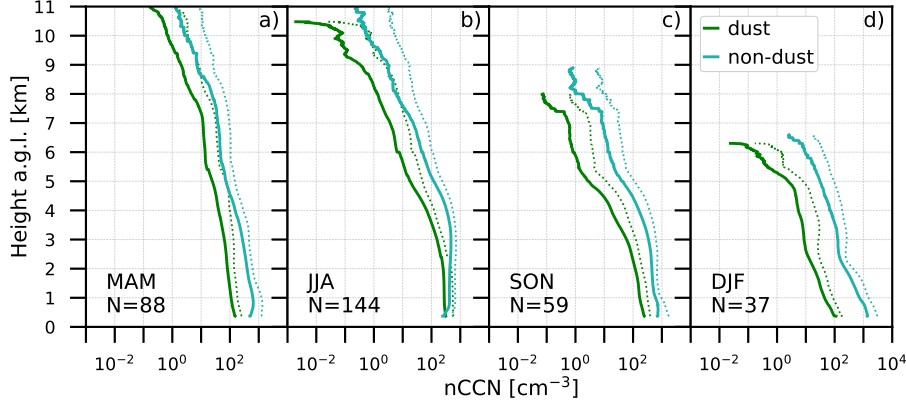

**Figure 15.** Same as Fig. 12, except for seasonal mean dust and non-dust CCN concentrations (for 0.2% supersaturation) in (a) spring, (b) summer, (c) autumn, and (d) winter. The dotted lines again show the (mean + SD) values. Considered nighttime observations N are given as numbers.





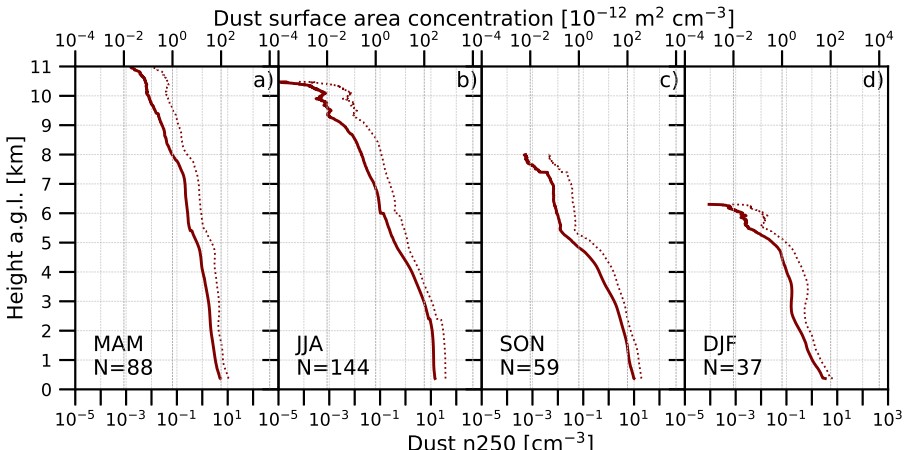

**Figure 16.** Same as Fig. 12, except for the seasonal mean dust particle number concentration $n_{250,d}$ (considering particles with radius >250 nm) and dust surface area concentration.

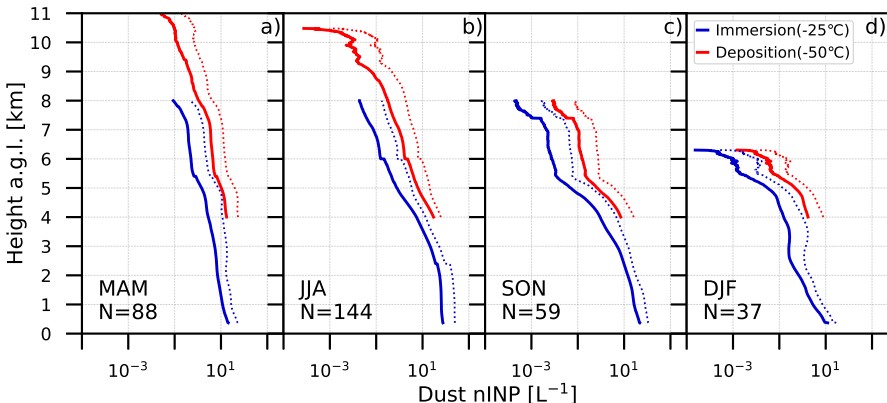

**Figure 17.** Same as Fig. 12, except for seasonal mean dust-related INP concentrations (nINP). The dotted lines show the (mean + SD) values. INP concentration is shown for fixed temperatures of $-25°C$ (blue, immersion freezing parameterization of DeMott et al. (2015) is used) and $-50°C$ (red, deposition nucleation parameterization of Ullrich et al. (2017) for an ice supersaturation value of 1.15).





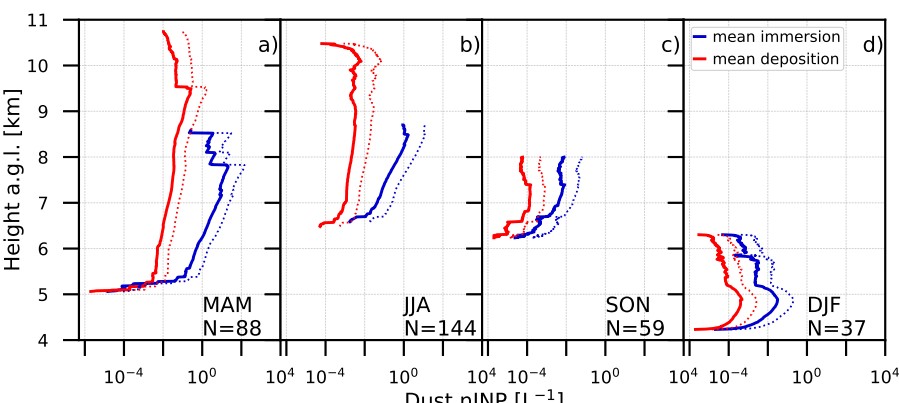

**Figure 18.** Same as Fig. 17, except considering actual temperature and pressure profiles.