# Peer review of "Long-term profiling of aerosol light-extinction, particle mass, cloud condensation nuclei, and ice-nucleating particle concentration over Dushanbe, Tajikistan, in Central Asia"

_Atmospheric Chemistry and Physics, 2019_

## Referee Comment (RC1) · Anonymous Referee #2 · 8 Jan 2020

Paper analyzes the results of 18 months lidar observations over Dushanbe and provide statistics for seasonal variation of aerosol parameters. Authors apply previously developed POLIPHON approach to derive profiles of CCN and INP. Manuscript is well and clearly written and can be published in ACP. I have just some technical comments.

Abstract, Ln. 1." . . .were conducted with a state-of-the-art multiwavelength lidar" Should be explained, which parameters make it "state-of-art"

p.4 Ln.11. "..of (30–40 sr) and for Central Asian aerosol pollution (30–50 sr)". References should be given. For dust, 30 sr is too low.

p.4 Ln.8 "...on typical particle linear depolarization ratio values (Müller et al., 2007; Tesche et al., 2009) for dust (0.31) and non-dust (0.05)". These are values for African dust. Are data for Asian dust available? What are values from CADEX measurements?

Are results for seasonal variations of lidar ratio, depolarization, Angstrom available?

P.8. Ln9 "Our findings are in reasonable agreement with the MODIS observations of the Ångström exponent" How reliable is Angstrom from MODIS over the bright surface? Definitely AERONET should be the primary instrument for comparison.

Fig.2. Does plot "d" shows particle depolarization? It's strange that it is close to zero at 5 km, where backscattering is also close to zero. How could authors get particle depolarization at such low backscattering?

Fig.6 needs some discussions. Lidar ratio (LR) at 532 is below 40 sr, which is low. Previously published values of lidar ratios should be reviewed. On this figure LR355>LR532, while during SAMUM the lidar ratios coincided. It should be discussed. Recall, that during SHADOW campaign in Africa, LR355 exceeded LR532, which was related to the spectral dependence of the dust imaginary part (Veselovskii, I., Goloub, P., Podvin, T., Bovchaliuk, V., Derimian, Y., Augustin, P., Fourmentin, M., Tanre, D., Korenskiy, M., Whiteman, D., Diallo, A., Ndiaye, T., Kolgotin, A., Dubovik, O.: Study of African dust with multi-wavelength Raman lidar during the "SHADOW" campaign in Senegal, Atm. Chem. Phys. 16, 7013–7028, 2016.) What is difference between African and Asian dust?

---

## Referee Comment (RC2) · Anonymous Referee #1 · 10 Jan 2020

This paper presents comprehensive analysis of the lidar observation in Dushanbe, Tajikistan in the Central Asian Dust Experiment (CADEX). The manuscript is well written in general. The analysis relies on the POLIPHONE method. Although references are suitably cited and accuracy estimates are given, it would be helpful if brief descriptions on the basic concept of the POLIPHONE method and assumptions involved in the method.

Specific comments: Table 1: It seems "bd" and "bc" in Table 1 are not explained. Figure 5: What is the reason for changing the trajectory arriving height with season? The

difference in trajectory may be due to the difference in the arriving height. Would it be possible to give more comprehensive presentation of the trajectory cluster analysis? Figure 13 and 14: What is the definition of the top height of dust layer? Figure 17 and 18: It is difficult to understand exactly the difference between these figures. If I understand correctly, the parameterization to estimate INP is dependent on temperature, but constant temperature (-25degC and -50degC for immersion and deposition nucleation) was used in Figure 17. In Figure 18, INP was derived for each lidar profile using actual temperature profile, and INP profiles were averaged later. Is that correct?

---

## Author Comment (AC1) · 19 Feb 2020

**General remarks**

We would like to thank the reviewers for their careful reading and helpful comments. Our answers to the reviewers' specific questions (in bold) are listed below. A suggested revised manuscript with the changes marked up in red is added at the end of this document. The main change is an additional table to elucidate the steps of the POLIPHON method. Furthermore, we clarified some of the used parameters (lidar ratios, conversion factors, temperature and pressure profiles from GDAS (2019)) We would like to mention that this manuscript is a first part of the CADEX results (climatology, environmental study, microphysical and cloud-relevant parameters). A second part will cover lidar-specific results (lidar and depolarization ratios, Ångström exponents) and will be submitted presumably in March this year (Hofer et al. (2020): Optical properties of Central Asian aerosol mixtures relevant for space lidar applications and aerosol typing at 355 and 532 nm).

**Answers to Reviewer 1**

**"...were conducted with a state-of-the-art multiwavelength lidar" Should be explained, which parameters make it "state-of-art"**
The observable parameters (particle backscatter coefficient at 355 nm, 532 nm, and 1064 nm, and particle linear depolarization ratio and particle extinction coefficient at 355 nm and 532 nm wavelength, and water vapor) as well as the directly derivable parameters (lidar ratio at 355 nm and 532 nm wavelength and backscatter-related and extinction-related Ångström exponents) make the system state-of-the-art.

**"...of (30–40 sr) and for Central Asian aerosol pollution (30–50 sr)". References should be given. For dust, 30 sr is too low.**
These ranges are taken from the Dushanbe values which will be published soon in Hofer et al. (2020). In the POLIPHON algorithm, 40 sr for dust an 50 sr for aerosol pollution have been used for the microphysical and cloud-relevant properties. For the DOT calculation in Fig. 8 and the seasonal mean extinction coefficients (red profiles in Fig. 10a–c), 35 sr have been used. We changed the sentence and state only the actually used values instead of the observed range. Furthermore, we cite Hofer et al. (2017) where we presented first cases studies with according values.

**"...on typical particle linear depolarization ratio values (...) for dust (0.31) and non-dust (0.05)". These are values for African dust. Are data for Asian dust available? What are values from CADEX measurements? Are results for seasonal variations of lidar ratio, depolarization, Ångström available?**
The same answer as above applies to this question. Yes, such results are available and will be published soon in Hofer et al. (2020). These mentioned depolarization ratio values are the used thresholds for the POLIPHON dust/non-dust separation. The applied depolarization ratio thresholds are assumed to be applicable to Central Asian dust as well. As stated in Ansmann et al. (2019), pure dust causes particle linear depolarization ratios of 0.3–0.35 around the world, disregarding the source region of the dust. Numerous studies and reviews (Tesche et al., 2009; Mamouri and Ansmann, 2014, 2017) and recent field campaigns (Groß

et al., 2015; Veselovskii et al., 2016; Haarig et al., 2017; Hofer et al., 2017) found the same result.

**"our findings are in reasonable agreement with the MODIS observations of the Ångström exponent" How reliable is Angstrom from MODIS over the bright surface? Definitely AERONET should be the primary instrument for comparison.**
In this section, we did not want to validate the MODIS values in any way. It is just a comparison to values presented in Rupakheti et al. (2019) which are in reasonable agreement to our measurements in that sense that a considerable part of the AOT is caused by anthropogenic fine-mode aerosol pollution which agrees with the high Ångström exponents observed by Rupakheti et al. (2019). Rupakheti et al. (2019) used Aqua-MODIS Collection 6.1 Deep Blue data. The Deep Blue algorithm is designed to retrieve AOT over bright surfaces (Hsu et al., 2004, 2013) but over land, it only provides AOT at three wavelengths. So, Rupakheti et al. (2019) used the AOT at 0.47 µm and 0.66 µm wavelength to calculate the Ångström exponent.
The sentence in brackets "(describing the AOT wavelength dependence in the spectral range from 440 to 870 nm)" is wrongly placed and misleading as it belongs to the AERONET Ångström exponent retrieval and not to MODIS. This is changed and the sentence is placed correctly two lines below.
Many other studies focus on the validation of MODIS retrievals using AERONET (e.g., Sayer et al., 2013). At this point, doing this with the Dushanbe AERONET data as well would be beyond the scope of this study. A comparison of lidar-derived to AERONET Ångström exponents for Dushanbe dust cases was shown in Hofer et al. (2019). In a follow-up study, more lidar-derived backscatter-related and extinction-related Ångström exponents will be presented (Hofer et al., 2020). Please also see Fig. C1.

**Fig. 2. Does plot "d" show particle depolarization? It's strange that it is close to zero at 5 km, where backscattering is also close to zero. How could authors get particle depolarization at such low backscattering?**
Between 5–5.5 km height, the backscatter coefficients are (mean $\pm$ SD) $19\pm19\cdot10^{-5}$ km$^{-1}$ sr$^{-1}$ at 355 nm, $3.6\pm4.6\cdot10^{-5}$ km$^{-1}$ sr$^{-1}$ at 532 nm, and $2.9\pm0.8\cdot10^{-5}$ km$^{-1}$ sr$^{-1}$ at 1064 nm wavelength. This is still higher than the reference value of $1\cdot10^{-5}$ km$^{-1}$ sr$^{-1}$ which has been placed at 10.5–12 km height. From the standard deviation though, it is visible how noisy the particle backscatter coefficients are in this low aerosol height range. The smooth particle depolarization ratio profile is a consequence of the strong vertical smoothing. In the proposed revised manuscript, the particle depolarization ratio profile in Fig. 2 is cut between 4.8–5.7 km and above 7.5 km.

**Fig. 6 needs some discussions. Lidar ratio (LR) at 532 is below 40 sr, which is low. Previously published values of lidar ratios should be reviewed. On this figure LR355>LR532, while during SAMUM the lidar ratios coincided. It should be discussed. Recall, that during SHADOW campaign in Africa, LR355 exceeded LR532, which was related to the spectral dependence of the dust imaginary part (Veselovskii et al., 2016). What is difference between African and Asian dust?**
Concerning this specific case, lidar (depolarization) ratios of $44\pm1$ sr ($0.21\pm0.01$) at 355 nm and $37\pm1$ sr ($0.29\pm0.01$) at 532 nm wavelength were measured (averaged between 1.5–2.5 km). The extinction-related Ångström exponent is $0.35\pm0.06$ and the AOT at 532 nm is 0.45.

Higher dust lidar ratios are a specific feature of West African dust (Tesche et al., 2011; Veselovskii et al., 2016), while Eastern Saharan and Middle Eastern dust again show lower lidar ratios (Schuster et al., 2012; Mamouri et al., 2013; Nisantzi et al., 2015). As stated above, the aim of this study is to present and describe the climatological results rather than the detailed lidar-specific aerosol optical parameters, which are subject of the soon to be published follow-up study Hofer et al. (2020). We generally measured higher lidar ratios at 355 nm than at 532 nm wavelength for most aerosol conditions during the CADEX campaign. A preview of the results of Hofer et al. (2020) is presented in Fig. C1, where low lidar ratios at elevated depolarization ratios are visible. During strong dust outbreaks though, the lidar ratios at both wavelengths are on average closer together. We would like not to speculate about the reason of this, particularly as it is not the scope of this study. A couple of studies investigated differences between West and East Saharan and Asian dust using a combination different remote sensing and modeling efforts, e.g., (Schuster et al., 2012; Su and Toon, 2011). It is not the scope of this study to answer this question, but in general, we consider lidar field campaigns, especially continuous, long-term measurements, together with in situ observations and laboratory and modeling studies as essential to get a better understanding of optical, radiative, microphysical, and mineralogical properties of (Central Asian) dust.

[Figure]

Figure C1: Scatter plots of layer-mean particle linear depolarization ratio against lidar ratio (a), extinction-related Ångström exponent against particle linear depolarization ratio (b), and against lidar ratio (c). Blue for 355 nm and green for 532 nm wavelength. The error bars denote the standard deviation of the mean in the respective averaging range (Hofer et al., 2020).

**Answers to Reviewer 2**

**Although references are suitably cited and accuracy estimates are given, it would be helpful if brief descriptions on the basic concept of the POLIPHONE method and assumptions involved in the method.**
We added the following Table (analogous to Mamouri and Ansmann (2016)) to make the sequential steps of the POLIPHON calculations clearer to the reader:

Table 1: Overview of the data analysis from the basic lidar-derived aerosol optical properties (particle backscatter and extinction coefficients, particle linear depolarization ratio) to the height profiles of CCN and INP concentrations. r denotes the particle radius.

| Step | Description |
| --- | --- |
| 1 | Retrieval of particle backscatter coefficient and particle linear depolarization ratio profiles at 532 nm wavelength |
| 2 | Separation of dust and non-dust backscatter coefficients using thresholds of the particle depolarization ratio for dust and non-dust |
| 3 | Conversion to dust and non-dust extinction coefficients from dust and non-dust particle backscatter coefficients using dust and non-dust lidar ratios |
| 4 | Conversion to dust and non-dust particle mass, number, and surface area concentrations from dust and non-dust extinction coefficient |
| 5 | Estimation of CCN concentration from dust (r>100 nm) and non-dust number concentration (r>50 nm) |
| 6 | Estimation of INP concentration using dust number concentration (r>250 nm) and temperature (immersion freezing), and using dust surface area concentration and temperature (deposition nucleation) |

**Table 1: It seems "bd" and "bc" in Table 1 are not explained.**

The multiplicative conversion factors denoted with capital $C$ are used for the calculation of, e.g., the number concentration of large dust particles $n_{250,\mathrm{d}}$ from the dust extinction coefficient $\alpha_\mathrm{d}$ as

$$n_{250,\mathrm{d}} = C_{\mathrm{n}_{250,\mathrm{d}}} \cdot \alpha_\mathrm{d} \,.$$

The exponents $b_\mathrm{d}$ and $b_\mathrm{c}$ for dust and continental haze (this also the reason why they do not have a unit) are used in addition for the calculation of the number concentrations including the smaller size fractions, $n_{100,\mathrm{d}}$ for dust and $n_{50,\mathrm{c}}$ for continental haze as

$$n_{100,\mathrm{d}} = C_{\mathrm{n}_{100,\mathrm{d}}} \cdot \alpha_\mathrm{d}^{b_\mathrm{d}}$$

and

$$n_{50,\mathrm{c}} = C_{\mathrm{n}_{60,\mathrm{c}}} \cdot \alpha_\mathrm{c}^{b_\mathrm{c}} \,.$$

In the revised manuscript, we added a short explanation in the text as follows: "For the $n_{100,\mathrm{d}}$ and $n_{50,\mathrm{c}}$ conversions, the exponents $b_\mathrm{d}$ and $b_\mathrm{c}$ are used."

**Figure 5: What is the reason for changing the trajectory arriving height with season? The difference in trajectory may be due to the difference in the arriving height. Would it be possible to give more comprehensive presentation of the trajectory cluster analysis?**

The decision to only show different arriving heights for different season was based on aerosol layer top heights presented in Fig. 3 and Tab. 2. Figures 5a and b show the clusters for the main aerosol layer, while Fig. 5c shows upper tropospheric air mass transport corresponding to the uppermost aerosol layers (blue stars in Fig. 3) which is basically valid for all seasons

but winter. For the sake of completeness, we like to present here all calculated HYSPLIT clusters for all seasons and heights (Fig. C2). The main argumentation (local/regional air masses in the main aerosol layer, Saharan/Middle Eastern air masses in the upper troposphere especially in spring and summer) is not impaired by this omission.

[Figure]

Figure C2: Matrix of all calculated HYSPLIT clusters for all season and heights. For spring (MAM, a–d), summer (JJA, e–h), autumn (SON, g–l), and winter (DJF, m–p). For 1.5 km (a,e,i,m) , 2.5 km (b,f,j,n), 4.5 km (c,g,k,o), and 7.5 km arrival height (d,h,l,p). d), f), and m) are shown in the manuscript in Fig. 5.

**Figure 13 and 14: What is the definition of the top height of dust layer?**
These top heights were determined by visual inspection, there is no single definition. By visual inspection of the profiles at very low vertical smoothing of 23 m, the following aerosol top heights were defined: The significant aerosol layer height was defined where the backscatter coefficient at 1064 nm wavelength reaches a first minimum in a range between about $0–1.5 \cdot 10^{-4}$ km$^{-1}$ sr$^{-1}$. If other aerosol layers at higher altitudes were present above the significant aerosol layer height, an uppermost layer height was defined where the backscatter coefficient decreases again to a range between about $0–5 \cdot 10^{-5}$ km$^{-1}$ sr$^{-1}$. This method is to a certain degree arbitrary. Therefore, more objective automatic layer detection methods were

applied as described in the manuscript, too. The agreement between the different results is good.

**Figure 17 and 18: It is difficult to understand exactly the difference between these figures.**

As you described it perfectly right, Fig. 17 shows the resulting values for applying the parametrizations on the mean dust number and surface concentration of all profiles at constant temperature of $-25°C$ and $-50°C$, respectively. Figure 18 shows the same calculation but based on the individual profiles at individual temperature and pressure conditions with afterward applied averaging. Indeed, it can be misleading to call these the "actually observed temperature and pressure" because temperature and pressure profiles from the GDAS model are used (GDAS, 2019). To clarify this and to make it more comprehensible we changed that in the manuscript as following: "As a final result, we present the INP concentration profiles for the found aerosol conditions in combination with the actual pressure and temperature profiles from GDAS (2019) in Fig. 18 to give an impression of typical (actual) INP concentration values for an ice supersaturation of 1.15."

**References in comments**

Ansmann, A., Mamouri, R.-E., Hofer, J., Baars, H., Althausen, D., and Abdullaev, S. F.: Dust mass, cloud condensation nuclei, and ice-nucleating particle profiling with polarization lidar: updated POLIPHON conversion factors from global AERONET analysis, Atmos. Meas. Tech., 12, 4849–4865, doi:10.5194/amt-12-4849-2019, 2019.

GDAS(2019): Global Data Assimilation System, meteorological data base, available at: https://www.ready.noaa.gov/gdas1.php, last access: 22 September, 2019.

Groß, S., Freudenthaler, V., Schepanski, K., Toledano, C., Schäfler, A., Ansmann, A., and Weinzierl, B.: Optical properties of long-range transported Saharan dust over Barbados as measured by dual-wavelength depolarization Raman lidar measurements, Atmos. Chem. Phys., 15, 11 067–11 080, doi:10.5194/acp-15-11067-2015, 2015.

Haarig, M., Ansmann, A., Althausen, D., Klepel, A., Groß, S., Freudenthaler, V., Toledano, C., Mamouri, R.-E., Farrell, D. A., Prescod, D. A., Marinou, E., Burton, S. P., Gasteiger, J., Engelmann, R., and Baars, H.: Triple-wavelength depolarization-ratio profiling of Saharan dust over Barbados during SALTRACE in 2013 and 2014, Atmos. Chem. Phys. Discuss., 2017, 1–43, doi:10.5194/acp-2017-170, 2017.

Hofer, J., Althausen, D., Abdullaev, S. F., Makhmudov, A. N., Nazarov, B. I., Schettler, G., Engelmann, R., Baars, H., Fomba, K. W., Müller, K., Heinold, B., Kandler, K., and Ansmann, A.: Long-term profiling of mineral dust and pollution aerosol with multiwavelength polarization Raman lidar at the Central Asian site of Dushanbe, Tajikistan: case studies, Atmos. Chem. Phys., 17, 14 559–14 577, doi:10.5194/acp-17-14559-2017, 2017.

Hofer, J., Althausen, D., Abdullaev, S. F., Makhmudov, A. N., Nazarov, B. I., Baars, H., Engelmann, R., and Ansmann, A.: Profiling aerosol optical properties at the Central Asian site of Dushanbe, Tajikistan: pure dust cases, in: Proceedings of the 29th International Laser Radar Conference (ILRC), 24–28 June 2019, Hefei, Anhui, China, pp. S2–180–S2–183, 2019.

Hofer, J., Ansmann, A., Althausen, D., Engelmann, R., Baars, H., Wandinger, U., Abdullaev, S. F., and Makhmudov, A. N.: Optical properties of Central Asian aerosol mixtures relevant for space lidar applications and aerosol typing at 355 and 532 nm, Atmos. Chem. Phys. Discuss, 2020, in preparation, 2020.

Hsu, N., Tsay, S.-C., King, M., and Herman, J. R.: Aerosol properties over bright-reflecting source regions, IEEE Trans. Geosci. Remote Sens., 42, 557–569, doi:10.1109/TGRS.2004.824067, 2004.

Hsu, N. C., Jeong, M.-J., Bettenhausen, C., Sayer, A. M., Hansell, R., Seftor, C. S., Huang, J., and Tsay, S.-C.: Enhanced Deep Blue aerosol retrieval algorithm: The second generation, J. Geophys. Res. Atmos., 118, 9296–9315, doi:10.1002/jgrd.50712, 2013.

Mamouri, R. E. and Ansmann, A.: Fine and coarse dust separation with polarization lidar, Atmos. Meas. Tech., 7, 3717–3735, doi:10.5194/amt-7-3717-2014, 2014.

Mamouri, R.-E. and Ansmann, A.: Potential of polarization lidar to provide profiles of CCN- and INP-relevant aerosol parameters, Atmos. Chem. Phys., 16, 5905–5931, doi:10.5194/acp-16-5905-2016, 2016.

Mamouri, R.-E. and Ansmann, A.: Potential of polarization/Raman lidar to separate fine dust, coarse dust, maritime, and anthropogenic aerosol profiles, Atmos. Meas. Tech., 10, 3403–3427, doi:10.5194/amt-10-3403-2017, 2017.

Mamouri, R. E., Ansmann, A., Nisantzi, A., Kokkalis, P., Schwarz, A., and Hadjimitsis, D. G.: Low Arabian dust extinction-to-backscatter ratio, Geophys. Res. Lett., 40, 4762–4766, doi:10.1002/grl.50898, 2013.

Nisantzi, A., Mamouri, R. E., Ansmann, A., Schuster, G. L., and Hadjimitsis, D. G.: Middle East versus Saharan dust extinction-to-backscatter ratios, Atmos. Chem. Phys., 15, 5203–5240, doi:10.5194/acp-15-7071-2015, 2015.

Rupakheti, D., Kang, S., Bilal, M., Gong, J., Xia, X., and Cong, Z.: Aerosol optical depth climatology over Central Asian countries based on Aqua-MODIS Collection 6.1 data: Aerosol variations and sources, Atmos. Env., 207, 205–214, doi:10.1016/j.atmosenv.2019.03.020, 2019.

Sayer, A. M., Hsu, N. C., Bettenhausen, C., and Jeong, M.-J.: Validation and uncertainty estimates for MODIS Collection 6 "Deep Blue" aerosol data, J. Geophys. Res. Atmos., 118, 7864–7872, doi:10.1002/jgrd.50600, 2013.

Schuster, G. L., Vaughan, M., MacDonnell, D., Su, W., Winker, D., Dubovik, O., Lapyonok, T., and Trepte, C.: Comparison of CALIPSO aerosol optical depth retrievals to AERONET measurements, and a climatology for the lidar ratio of dust, Atmos. Chem. Phys., 12, 7431–7452, doi:10.5194/acp-12-7431-2012, 2012.

Su, L. and Toon, O. B.: Saharan and Asian dust: similarities and differences determined by CALIPSO, AERONET, and a coupled climate-aerosol microphysical model, Atmos. Chem. Phys., 11, 3263–3280, doi:10.5194/acp-11-3263-2011, 2011.

Tesche, M., Ansmann, A., Müller, D., Althausen, D., Engelmann, R., Freudenthaler, V., and Groß, S.: Vertically resolved separation of dust and smoke over Cape Verde by using multiwavelength Raman and polarization lidars during Saharan Mineral Dust Experiment 2008, J. Geophys. Res. Atmos., 114, D13 202, doi:10.1029/2009JD011862, 2009.

Tesche, M., Groß, S., Ansmann, A., Müller, D., Althausen, D., Freudenthaler, V., and Esselborn, M.: Profiling of Saharan dust and biomass-burning smoke with multiwavelength polarization Raman lidar at Cape Verde, Tellus B, 63, 649–676, doi:10.1111/j.1600-0889.2011.00548.x, 2011.

Veselovskii, I., Goloub, P., Podvin, T., Bovchaliuk, V., Derimian, Y., Augustin, P., Fourmentin, M., Tanre, D., Korenskiy, M., Whiteman, D. N., Diallo, A., Ndiaye, T., Kolgotin, A., and Dubovik, O.: Retrieval of optical and physical properties of African dust from multiwavelength Raman lidar measurements during the SHADOW campaign in Senegal, Atmos. Chem. Phys., 16, 7013–7028, doi:10.5194/acp-16-7013-2016, 2016.

[revised manuscript text omitted]